# KCTD15 Protein Expression in Peripheral Blood and Acute Myeloid Leukemia

**DOI:** 10.3390/diagnostics10060371

**Published:** 2020-06-04

**Authors:** Giovanni Smaldone, Luigi Coppola, Mariarosaria Incoronato, Rosanna Parasole, Mimmo Ripaldi, Luigi Vitagliano, Peppino Mirabelli, Marco Salvatore

**Affiliations:** 1IRCCS SDN, Napoli, Via E. Gianturco 113, 80143 Naples, Italy; giovanni.smaldone@synlab.it (G.S.); lcoppola@sdn-napoli.it (L.C.); mincoronato@sdn-napoli.it (M.I.); direzionescientifica@sdn-napoli.it (M.S.); 2Department of Pediatric Hematology-Oncology, Santobono-Pausilipon Hospital, 80129 Naples, Italy; rparasol64@gmail.com (R.P.); mimmo.ripaldi@tin.it (M.R.); 3Institute of Biostructures and Bioimaging, C.N.R., 80134 Napoli, Italy

**Keywords:** KCTD15, peripheral blood, myeloid cell lines, flow cytometry, biomarker, diagnostics

## Abstract

Leukocytes are major cellular components of the inflammatory and immune response systems. After their generation in the bone marrow from hematopoietic stem cells, they maturate as granulocytes (neutrophils, eosinophils, and basophils), monocytes, and lymphocytes. The abnormal accumulation and proliferation of immature blood cells (blasts) lead to severe and widespread diseases such as leukemia. We have recently shown that KCTD15, a member of the potassium channel tetramerization domain containing protein family (KCTD), is remarkably upregulated in leukemic B-cells. Here, we extend our investigation by monitoring the KCTD15 expression levels in circulating lymphocytes, monocytes, and granulocytes, as well as in leukemia cells. Significant differences in the expression level of KCTD15 were detected in normal lymphocytes, monocytes, and granulocytes. Interestingly, we also found overexpression of the protein following leukemic transformation in the case of myeloid cell lineage. Indeed, KCTD15 was found to be upregulated in K562 and NB4 cells, as well as in HL-60 cell lines. This in vitro finding was corroborated by the analysis of KCTD15 mRNA of acute myeloid leukemia (AML) patients reported in the Microarray Innovations in Leukemia (MILE) dataset. Collectively, the present data open interesting perspectives for understanding the maturation process of leukocytes and for the diagnosis/therapy of acute leukemias.

## 1. Introduction

White blood cells, also denoted as leukocytes, are the major cellular components of the inflammatory and immune response systems that protect against foreign invaders and neoplasia, and they are also involved in the repair of damaged tissue [1,2]. Leukocytes originate in the bone marrow from multipotent cells known as hematopoietic stem cells. The differentiation/maturation of these cells generates different types of leukocytes [3,4]. Depending on their functional and morphological features, white blood cells can be categorized into granulocytes (neutrophils, eosinophils, and basophils), monocytes, and lymphocytes (T-cells, B-cells, and NK cells). These cells display analogies and differences in the expression of specific cell surface receptors. Indeed, lymphocytes, monocytes, and granulocytes can be effectively identified and distinguished by monitoring the expression level of the CD45 antigen (also defined as leukocyte common antigen, a membrane glycoprotein expressed on almost all hematopoietic cells except for mature erythrocytes) and by analyzing their light-scattering properties using flow cytometry (FCM) [5,6,7]. Moreover, additional antigens, such as CD14 and CD33, are used to better define the monocyte subsets, whereas CD66b expression levels are used for clustering granulocytes [8,9].

Leukemia is a heterogeneous and multifactorial blood cancer characterized by an abnormal accumulation of blood cells that are not fully developed (blasts) [10]. Depending on the time evolution of the disease and on the nature of blasts, leukemias are generally subdivided into acute lymphoblastic leukemia (ALL), acute myeloid leukemia (AML), chronic lymphocytic leukemia (CLL), and chronic myeloid leukemia (CML) [10,11]. CLL and AML are the most frequent leukemias in adults, whereas ALL is the most common cancer diagnosed in children. Acute myeloid leukemia (AML) is the most common acute leukemia in adults. According to the World Health Organization Classification of Tumours of Haematopoietic and Lymphoid Tissues [10], the diagnostic procedures used to identify leukemic cells and classify acute leukemias are based on (i) morphologic assessment of bone marrow specimens and blood smears, (ii) analysis of the expression of cell-surface or cytoplasmic markers utilizing flow cytometry, (iii) identification of chromosomal findings through conventional cytogenetic testing, and, more recently, (iv) screening for selected molecular genetic lesions.

The identification of biomarkers and key players in the etiology of acute leukemias is fundamental for a better understanding of the molecular basis of these diseases and the setup of optimal strategies for their diagnosis, prognosis, treatment, and monitoring. We have recently discovered that KCTD15, a member of the potassium channel tetramerization domain (KCTD) protein family [12,13], is remarkably overexpressed in both B-cell ALL blasts and cell lines [14]. Notably, KCTD15 activity is critical for B-cell proliferation since its silencing induces the arrest of cellular proliferation and apoptosis. Moreover, the proliferation of lymphocytes induced in vitro by PWM (poke weed mitogen) stimulation is associated with KCTD15 upregulation, thus confirming the importance of this protein for sustaining the growth of the lymphoid cells.

In this scenario, using a multiparametric flow cytometry approach, we here evaluate KCTD15 expression levels in circulating lymphocytes, monocytes, and granulocytes, as well as in human acute myeloid leukemia-derived cell lines. The comparison of KCTD15 expression levels detected in these cells with those observed in the peripheral blood (PB) of healthy subjects demonstrates the upregulation of this protein in both the physiological and pathological evolution of white blood cells.

## 2. Results

To gain insights into the expression profiles of KCTD15 in different conditions, we preliminarily monitored the physiological protein levels in lymphocytes, monocytes, and granulocytes in PB (Section 3.1). We then selected some representative AML cell lines (Section 3.2). These results were extended and validated by exploiting the data reported in the Microarray Innovations in Leukemia (MILE) study (Section 3.3).

### 3.1. Intracellular KCTD15 Expression in Peripheral Blood

The analysis of the KCTD15 intensity of expression in PB white blood cells was performed using a flow cytometry approach. To achieve this goal, PB samples were stained for CD45 (a pan-leucocyte antigen) and CD14 (a monocyte-specific marker), as well as for intracellular KCTD15. The gating strategy was performed as follow: single-cell events were selected on an FSC-H (forward scatter height) vs. FSC-A (forward scatter area) dot plot (Figure 1a), then fixed single live cells were identified on an FSC-A vs. SSC-A (side scatter area) dot plot (Figure 1b). The CD45 vs. SSC dot plot was used to identify lymphocytes (Figure 1c; blue events), monocytes (Figure 1c; orange events), and granulocytes (Figure 1c; red events), according to the different expression levels of the CD45 antigen and light SSC properties. The CD14 vs. SSC dot plot (Figure 1d) was done to evaluate the possible occurrence of monocytes contamination in the lymphocytes and/or granulocytes gates previously applied. This gating strategy allowed us to evaluate KCTD15 expression intensity in lymphocytes, monocytes, and granulocytes, as illustrated in the overlay histogram reported (Figure 1e). A global analysis of the results indicates that KCTD15 is clearly expressed in all of the leukocytes considered here. Nevertheless, KCTD15 expression is significantly higher in the myeloid compartment compared to circulating lymphoid cells (Figure 1f; Table 1). Moreover, significant differences have also been detected between granulocytes and monocytes, with the former showing larger expression levels of KCTD15. The flow cytometry results shown in Figure 1a–e were derived from a single subject (Subject #3) and fully confirmed by the inspection of PB samples obtained from 13 additional healthy individuals (Appendix A, *n* = 14). The mean KCTD15 intensity of expression for the selected WBC subpopulations is reported in Appendix A for each subject.

### 3.2. KCTD15 Protein Expression in Human Acute Myeloid Cell Lines

The analysis reported in the previous paragraph and our previous observation of the upregulation of KCTD15 in ALL cell lines and patients [14] prompted us to evaluate the expression levels of the protein in acute myeloid leukemia (AML)-derived cell lines. To this aim, we selected three model systems (NB4, K562, and HL60) widely used for AML in vitro study as they are featured by a different grade of differentiation and peculiar chromosomal translocations. In particular, (i) NB4 is a human acute promyelocytic leukemia-derived cell line characterized by the t(15;17) PML-RARA fusion gene, (ii) HL60 are cells derived from a human acute myeloid leukemia classified as FAB M2 and featured by MYC oncogene amplification, and (iii) K562 cells are obtained from chronic myeloid leukemia (CML) in blast crisis and the K562 cells carry the t(9;22) BCR-ABL1 e14-a2 (b3-a2) fusion gene. The KCTD15 levels of these cells were compared with those detected in peripheral blood mononuclear cells (PBMC), as defined by FCM, an analysis that contains lymphocytes and monocytes. The FCM analysis presented in Figure 2a,b and data on KCTD15 FMI (mean fluorescence intensity), reported in Table 1, clearly demonstrate that the protein is remarkably upregulated in the AML cell lines. This observation is fully corroborated by the Western blot analysis performed on the same systems (Figure 2c).

To further validate the potential of KCTD15 expression levels as a tool for distinguishing leukemic myeloid cells from normal PB-MNC(Pheripheral Blood MonoNuclear Cells), we labeled HL-60 cells with Violet-Cell Tracer dye. This strategy allowed us to simulate an AML sample by mixing fluorescent-violet labeled HL60 cells into normal whole blood. Thanks to this approach, KCTD15 expression intensity was measured simultaneously in HL60 as well as in white blood cells. As shown in Figure 3, KCTD15 expression is increased in HL60 cells in comparison to all cell types residing in peripheral blood.

### 3.3. KCTD15 mRNA Expression in AML and Healthy BM

The observation that KCTD15 is upregulated in AML cell lines prompted us to interrogate the Microarray Innovations in Leukemia (MILE) study dataset (accession number GEO13159), looking for KCTD15 mRNA levels. This dataset is available online through the Bloodspot website (available online: http://servers.binf.ku.dk/bloodspot/, accessed on 20 April 2020) [15] and allows the download of the data of interest for each mRNA included in the microarray study. Exploiting these data, we were able to evaluate the KCTD15 mRNA levels, as shown in Figure 4. KCTD15 mRNA was significantly upregulated (one-way ANOVA statistical analysis) in AML samples in comparison to healthy BM cells. In particular, the highest KCTD15 mRNA expression levels were detected in the case of AML with t(15;17) translocation that leads to the PML-RAR-alpha fusion protein expression. This finding fully corroborates our in vitro study regarding the increased KCTD15 expression found in the NB4 cell line.

## 3. Discussion

Recently, we identified KCTD15 as a novel player involved in the pathophysiology of pediatric acute lymphoid leukemias [14]. We showed that KCTD15 was strongly upregulated at mRNA and protein levels in B-ALL samples and cell lines in comparison to peripheral blood mononuclear cells, as well as bone marrow cells after antileukemic therapy.

Here, we decided to better describe the pattern of KCTD15 expression in peripheral blood cells as well as in acute myeloid leukemia cell lines and samples. In particular, to study the KCTD15 expression in lymphocytes, monocytes, and granulocytes, we decided to adopt a multiparametric FCM approach. In general, FCM has been used by clinical and experimental hematologists to identify distinct cell types based on extracellular or surface marker expression; however, it can be successfully used for the study of intracellular markers and complex signaling events, especially in the case of low-expressed protein antigens [16,17]. Therefore, the intracellular expression levels of KCTD15 was determined in association with surface CD45 and CD14 antigens to correctly discriminate lymphocytes (CD45bright/SSClow/CD14Neg), monocytes (CD45dim/SSCdim/CD14pos), and granulocytes (CD45dim/SSCbrigh/CD14Neg) [16,17]. In the present study, normal resting lymphocytes disclosed very low expression levels of KCTD15 protein in comparison to the intermediate expression levels detected in monocytes and granulocytes. This finding is in line with our previous report, where we showed that the KCTD15 basal levels increased following in vitro stimulation with pokeweed mitogen (PWM). The role of KCTD15 in lymphocytes is still not clear and needs to be further studied; however, it is important to consider experimental observations reported by other authors for other KCTD family members, i.e., KCTD5 and KCTD9. In particular, Bayón et al. [18] reported that KCTD5, like KCTD15, has low expression levels in peripheral blood lymphocytes, and its expression profile can be upregulated after in vitro stimulation with 4β-phorbol 12-myristate 13-acetate (PMA) plus ionomycin. In the case of KCTD9, Zhang X et al. [19] showed that this protein was important for NK cell activation in mouse models as NK cells reduced their cytokine production and cytotoxicity upon KCTD9 deletion.

As regards the myeloid compartment, KCTD15 levels progressively increased in monocytes and granulocytes when compared to circulating resting lymphocytes. To the best of our knowledge, this observation is new since KCTD proteins have never been studied in granulocytes as well as monocytes. To extend our study, we decided to investigate KCTD15 expression in myeloid cell lines with different levels of differentiation [20]. Indeed, HL60 is an immature FAB-M2-derived cell line with the ability to differentiate the monocyte/macrophage lineage upon stimulation with PMA [21]. NB4 is a model system derived from acute promyelocytic leukemia and it can differentiate granulocytes after stimulation with all-trans-retinoic acid [22]. Finally, K562 is a cell line derived from chronic myeloid leukemia in blast crisis and it is able to differentiate the erythroid lineage following induction with hemin [23]. In all the cell lines included, we were able to identify a strong upregulation of KCTD15 levels in comparison to those observed in normal white blood cells. To validate these in vitro findings, we exploited the online available MILE study dataset for studying KCTD15 mRNA expression levels in AML subsets, classified according to their cytogenetic alterations [10]. KCTD15 levels were upregulated in all AML cases in comparison to those found in the healthy bone marrow. Interestingly, the highest levels of KCTD15 mRNA were detectable in the case of AML with t(15;17), in line with our in vitro observations on the NB4 cell line, which is a model system derived from the same AML subtype.

These observations, along with our previous findings on the remarkable upregulation of the protein in ALL neoplastic transformation, strongly suggest that KCTD15 could be an important player involved in the etiology of different types of leukemia. The pathophysiology of acute leukemia is complex, and many experimental observations highlighted that, especially in the case of AML, the alterations at the cellular and molecular levels are determinants for disease occurrence [24]. Currently, cytogenetic markers are the most important for risk stratification and treatment of AML patients. However, with the advent of new technologies, the detection of other molecular markers, such as point mutations and characterization of epigenetic and proteomic profiles, have begun to play an important role in how the disease is approached [25]. Therefore, the identification of new AML biomarkers contributes to a better understanding of the molecular basis of the disease, especially for screening, diagnosis, prognosis, and monitoring of AML, as well as the possibility of predicting each individual’s response to treatment.

The precise role of KCTD15 in sustaining leukemic cell growth is still not understood; however, it is important to consider the biochemical features of KCTD15. Indeed, KCTD15 belongs to the KCTD family whose founding feature is the presence of a BTB (broad-complex, tramtrack, and bric-à-brac) domain in all members of the family [12,13]. This BTB domain, which is located in the N-terminal region of these proteins, is associated with C-terminal regions that are generally different for different members of the family, although subclasses of KCTD proteins sharing similarities in the entire sequence have been identified and classified (clades) [12]. One puzzling feature of these proteins is their involvement in a countless number of frequently unrelated, physiopathological processes. This include proliferation, differentiation, apoptosis, and metabolism [12,13,26], whereas improper regulation of KCTD genes has been associated with various diseases, including medulloblastoma [27], breast carcinoma, epilepsy [28], autism [29], rare genetic diseases [30,31], leukemia [14], obesity [32,33,34], and pulmonary inflammation [26]. Although this observation may be partially explained, taking into account the diversity of the C-terminal regions of KCTD, it is intriguing that members of the same clades or even the same proteins have been associated with different and unrelated pathological states. This is the case of the clade constituted by the highly homologous members KCTD1/KCTD15 [35]. Indeed, these highly homologous proteins play important roles in obesity [32,33,34], cancers [14,36], and scalp–ear–nipple syndrome [30,31]. Although KCTD1/KCTD15 biochemical function is largely unknown, also due to the lack of comprehensive structural data on these proteins [37,38], the above observations prompt for a role in the regulation of some basic process of cell life.

In conclusion, here, we show that KCTD15 plays an active role in both the physiology and pathological transformations of leukocytes. It can be surmised that uncontrolled regulation of the physiological KCTD15 expression in cells of both the myeloid and lymphoid compartments may be an important factor that favors the insurgence of the disease. Although the biochemical aspects of these activities are yet to be unraveled, present findings open new scenarios for the diagnosis and the therapy of these severe and widespread diseases.

## 4. Materials and Methods

Peripheral blood mononuclear cells (PBMCs) and cell lines.

Peripheral blood (PB) samples for experimental purposes were obtained from 14 healthy adult subjects (median age 39 years, 7 males and 7 females) and collected in 3 mL EDTA vacutainer tubes (Becton Dickinson, CA, USA, Catalog. #367835). All participants provided informed consent according to the study and experimental protocols approved by the local ethical committee (Comitato Etico IRCCS Pascale, Naples, Italy) of IRCCS-SDN with protocol number 6/16 of 14/09/2016 and by the local ethical committee of AORN Santobono-Pausilipon (Comitato Etico Cardarelli/Pausilion, Naples, Italy) with number 94 of 08/02/2017, following relevant guidelines and regulations.

White blood cells (WBC) were derived by PB dilution 1:20 with VersaLyse Solution (A09777, Beckman-Coulter, Brea, CA, USA) for red blood cell removal by osmotic shock. Peripheral blood mononuclear cells (PBMC) were obtained by centrifugation on density gradient media (Pancoll^®^ density 1077 g/L, PanBiotech, Aidenbach, Germany) at 400× *g* for 30 min.

The following authenticated human cell lines were used: HL-60, K562, and NB-4. Culture media (Sigma-Aldrich, St. Louis, MO, USA) was composed of Iscove’s modified Dulbecco’s medium supplemented with 2 mmol/L L-glutamine (Sigma-Aldrich) and 10% heat-inactivated FBS (ThermoFisher, GIBCO). All cell lines were cultured at 37 °C in a humidified atmosphere with 5% CO_2_.

The biological samples and cell lines included in this study were provided and processed by the Biobank of the SDN institute [39,40].

### 4.1. Flow Cytometry Experiments

For the flow cytometry experiments describing the KCTD15 expression on PB cells, we used the Cytoflex V2-B4-R2 (Beckman-Coulter, Brea, CA, USA). Conversely, for KCTD15 expression in HL-60, K562, and NB-4 cell lines as well as PBMC we used Cytomics FC500 (Beckman-Coulter, Brea, CA, USA) cytofluorimeters. QC fluorospheres were used before each experiment for verification of the flow cytometer’s optical alignment and fluidics system.

Intracellular or combined intracellular plus surface staining was performed by the use of the PerFix Expose kit (B26976, Beckman Coulter, Brea, CA, USA) according to the manufacturer’s instructions. Briefly, it consists of three ready-to-use reagents and a final solution requiring a 20-fold dilution before use. This kit was used to prepare biological samples for analysis of intracellular determinants by flow cytometry (FCM) through the enhancement of the signal-to-noise ratio of most intracellular antigens, including phosphor-epitopes. In addition, it allows the detection of several surface antigens together with intracellular markers. In the present study, we exploited the PerFix Expose kit for the detection of KCTD15 intracellular expression in peripheral blood cell samples from healthy subjects and AML cell lines. Purified PB cells and AML cells were fixed and permeabilized using Buffer 1 and 2, according to the manufacturer’s instructions. Then, antibody staining was performed for 20 min in Buffer 3 reagent using an unconjugated anti KCTD15 monoclonal antibody (GTX50002, Genetex International, USA). After 2 wash steps, the cells were incubated in Buffer 3 with an anti-mouse secondary FITC conjugated antibody (ab7064, Abcam, UK). For multiparameter FCM experiments where surface and intracellular antigen expression was contemporarily evaluated, the incubation with directly conjugated anti-CD45-KO or PC5 (B36294, Beckman Coulter) and anti-CD14-PC5-5 or PE (A70204, Beckman Coulter) antibodies was performed after incubation with fluorescent antimouse to prevent cross-species reactivity.

HL60 cells staining with CellTrace™ Violet was performed before the onset of FCM protocol, according to the manufacturer’s instructions. Briefly, actively growing HL60 cells (2 × 10^6^ cells) were diluted in 1 mL DPBS (Invitrogen Life Science) containing 5 μM of CellTrace™ Violet (Invitrogen, Cat. No. C34557) and incubated for 20 min at RT. Then, 5 mL of DPBS supplemented with 2% FBS was added to the initial 1 mL volume. After centrifugation, stained HL60 cells were mixed up in 100 μL of fresh PB at a concentration of 20,000 cells/μL in whole blood.

### 4.2. Western Blot Assays

Lysates from human AML cell lines and PBMC (50 µg of protein extracts) were analyzed by Western blot to check the expression of the protein. Antibodies used were anti KCTD15 (GTX50002, Genetex International, USA) and anti β-actin (ab11004, Abcam, UK) as internal controls. Proteins were acquired using the ChemiDoc imaging system (Bio-Rad, USA) coupled with Image Lab software.

### 4.3. Statistical Analysis and Reproducibility

*p*-values were calculated using Graphpad Prism 7 (Graphpad Software, Graphpad Holdings, LLC., CA, USA). Numbers of biological and/or technical replicates, as well as a description of the statistical parameters, are stated in the figure legends. All experimental images are representative of at least two independent experiments.

## Figures and Tables

**Figure 1 diagnostics-10-00371-f001:**
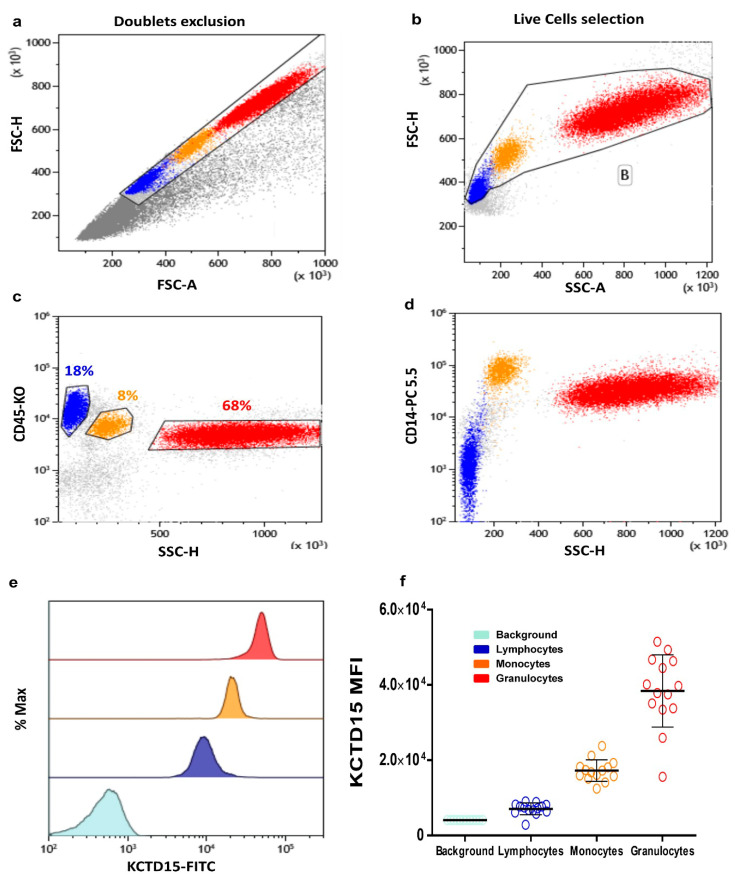
Potassium channel tetramerization domain (KCTD)15 expression in peripheral blood cells. (**a**) Forward scatter (FSC)-Height vs. FSC-Area dot plot used for the selection of single cells and doublets exclusion. (**b**) FSC vs. side scatter area (SSC) dot plot for the selection of single live cells. (**c**) CD45-KO vs. SSC color plot shows the identification of the lymphocytes (blue), monocytes (orange), and granulocytes (red) based on CD45 intensity of expression and light side scatter. (**d**) CD14-PC 5.5 vs. SSC density plot shows the expression of CD14 for better identification of the monocytes. (**e**) Overlay histogram showing the increased KCTD15 fluorescence intensity in granulocytes (red), monocytes (orange), and lymphocytes (blue) when compared to II-FITC antibody-labeled peripheral blood (light blue). (**f**) Scatter plot displaying the mean value of KCTD15 fluorescence intensity with standard deviations (SD) for the background (turkey circles; mean = 4089, SD = 1814), for lymphocytes (blue circles; mean = 7030, SD = 1572), for monocytes (orange circles; mean = 17,223, SD = 2863) and granulocytes (red circles; mean = 38.371, SD = 9597), as derived from the 14 cases studied here. The differences of KCTD15 fluorescence intensity of lymphocytes, monocytes, and granulocytes, compared to the background, are highly significant (*p* < 0.001, unpaired *t*-tests in all cases). Moreover, the difference detected for granulocytes vs. lymphocytes, granulocytes vs. monocytes, and lymphocytes vs. monocytes are also significant (*p* < 0.001, unpaired *t*-tests in all cases).

**Figure 2 diagnostics-10-00371-f002:**
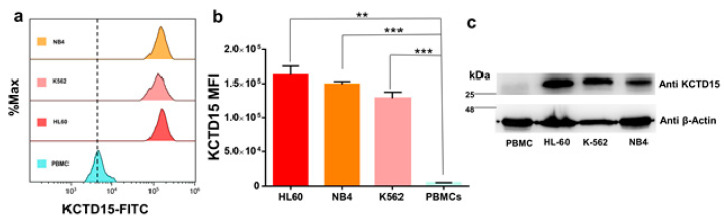
KCTD15 expression in the acute myeloid leukemia (AML) cell lines. (**a**) Overlay histogram showing the increased KCTD15 fluorescence intensity in NB-4, K562, and HL-60 cell lines when compared to peripheral blood mononuclear cells (PBMC). Histograms are normalized as a percentage of maximum count value (% Max) on the vertical axis. (**b**) Bar-plot displaying KCTD15 fluorescence intensity (in terms of mean of fluorescence and SD) in HL-60 (red), NB-4 (orange), and K562 (pink) compared to PBMC (light blue). ** *p* < 0.01, **** *p* < 0.0001, unpaired *t*-test. The error bar represents SD. (**c**) Western blot of KCTD15 protein in human AML in vitro model systems (HL-60, NB-4, and K-562), as well as PBMC. The number represents the molecular weight of the protein marker expressed in kDa. Experiments displayed in panels a–c were repeated twice with similar results.

**Figure 3 diagnostics-10-00371-f003:**
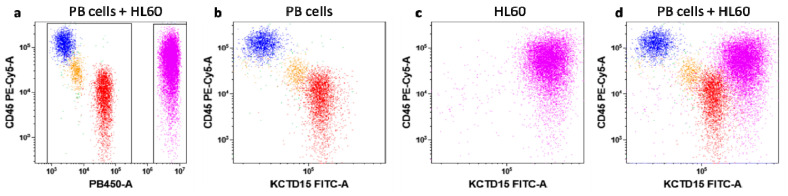
KCTD15 expression in peripheral blood (PB) and HL60 cells. Before the flow cytometry (FCM) acquisition of dot-plots (**a**)–(**d**), HL60 cells were stained with Violet Cell Tracer. Dot-plot (**a**) shows HL60 (violet events; right gate) distinguished from peripheral blood cellular populations (PB-cells; blue lymphocytes, orange monocytes, red granulocytes; left gate) due to Violet Cell Tracer fluorescence detected in PB450 channel. Dot-plots (**b**) and (**c**) display the surface CD45 vs. intracellular KCTD15 expression in PB-cells and HL60, respectively. Dot-plot (**d**) shows the brighter KCTD15 expression in HL60 cells in comparison to PB cell subsets.

**Figure 4 diagnostics-10-00371-f004:**
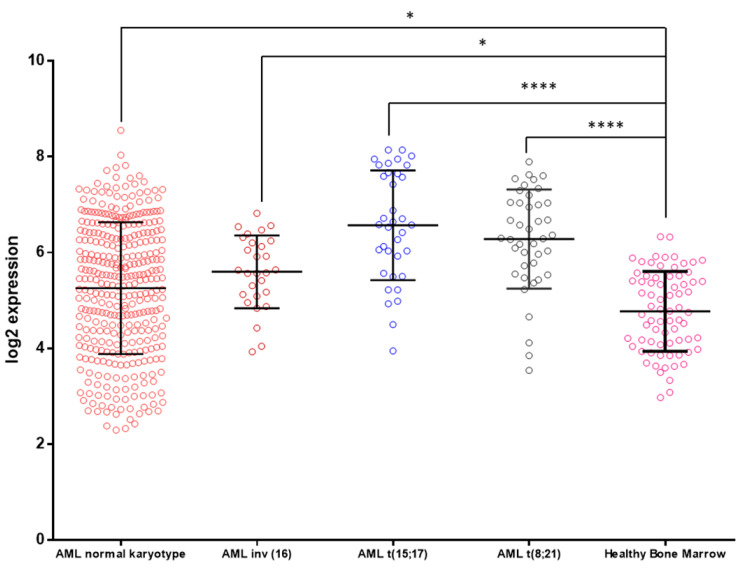
KCTD15 expression in human AML samples. Microarray data of KCTD15 were obtained from the MILE study dataset (accession number GEO13159) and plotted as log2 transformed expression values. AML samples were classified as AML normal karyotype (mean = 5.26, SD = 1.38, *n*  =  351 samples, red circles), AML inv (16) (mean = 5.60, SD = 0.75, *n*  =  28 samples, Bordeaux circles), AML t(15;17) (mean = 6.57, SD = 1.14, *n*  =  36 samples, blue circles), AML t(8;21) (mean = 6.28, SD = 1.03, *n*  =  41 samples, black circles), and healthy bone marrow (mean = 4.77, SD = 0.83, *n*  =  73 samples, pink circles). One-way ANOVA with Bonferroni’s multiple comparison test was applied; **** *p*  <  0.0001, * *p*  <  0.05.

**Table 1 diagnostics-10-00371-t001:** KCTD15 intensity of expression (mean +/- SD) in circulating HL-60, NB-4, and K562 AML cell lines and PBMC.

HL-60		NB-4		K562		PBMC	
Mean	SD	Mean	SD	Mean	SD	Mean	SD
162,296	±13,303	148,944	±4350	128,557	±8971	4622	±470.6

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
