# Peer review of "KCTD15 Protein Expression in Peripheral Blood and Acute Myeloid Leukemia"

_diagnostics, 2020, doi:10.3390/diagnostics10060371_

Round 1

Reviewer 1 Report

In this study, Smaldone and colleagues report elevated KCTD15 expression in peripheral blood and acute myeloid leukaemia cell lines. These findings may be interest, however the data presented was generated using only four peripheral blood samples and three different leukaemia cell lines and as such represent only preliminary results which should be confirmed on a larger cohort of patient samples and larger panel of cell lines. The statement of the authors on the possible role of the KCTD15 ('KCTD15 plays an active role in both physiology and pathological transformations of leukocytes') is not supported by the data presented in the manuscript. The paper is unfortunately not eligible for publication in Diagnostics.

Author Response

Response to Reviewer 1 Comments

In this study, Smaldone and colleagues report elevated KCTD15 expression in peripheral blood and acute myeloid leukemia cell lines. These findings may be interest, however the data presented was generated using only four peripheral blood samples and three different leukemia cell lines and as such represent only preliminary results which should be confirmed on a larger cohort of patient samples and larger panel of cell lines. The statement of the authors on the possible role of the KCTD15 ('KCTD15 plays an active role in both physiology and pathological transformations of leukocytes') is not supported by the data presented in the manuscript. The paper is unfortunately not eligible for publication in Diagnostics.

Response

We thank the referee for reviewing this manuscript. Taking in account the criticism of the reviewer #1 and the observation of the other reviewers, we performed additional analyses and experiments to corroborate and expand the conclusions of the original manuscript. To ameliorate our paper we decided to:

-              increase the number of PB samples tested for KCTD15 expression (14 patients). See figure 1

-              extend our analysis including the AML and BM patient cohort described in the MILE study dataset. See figure 4. In our opinion, the analysis that was made on mRNA expression level derived from AML patients demonstrates that the finding obtained on cell lines can be extended to patients.

-              improve the overall clarity of the manuscript and extend in the discussion the implications of the findings. 

Reviewer 2 Report

In this paper, the authors consider expression levels of KCTD15 on various types, building upon a previous Scientific Reports paper from 2019. It extends investigation of KCTD15 expression to lymphocytes, monocytes and granulocytes. It also finds KCTD15 expression is increased in several AML model cell lines. These findings are in line with the previously discovered role of KCTD15 in ALL.

Overall, this paper represents an incremental advance over the previously published work. KCTD15 levels are found to be upregulated in several new cell types, as well as several new oncogenic mutant cell lines. Its findings may be of interest to those in the field.

Scientifically, the paper requires better description and analysis of the results, as outlined specifically below. Otherwise, methods are well described and the results are clearly presented. No new experiments need be performed. The paper should be accepted once the problems below are addressed.

Section 3.1. Intracellular KCTD15 expression in peripheral blood

The purpose of the bar graph in Figure 1 is somewhat unclear – is the point that KCTD15 levels should be compare between different patients, or between cell types? If the latter, then perhaps plotting all four patients together would be more informative.

In supplementary figures 2(e) and 3(e), the KCTD15 expression histogram displays a bimodal distribution for Lymphocytes. Please comment on why a large subset of Lymphocytes appear to have a much lower expression of KCTD15, but only in some patients.

Section 3.2. KCTD15 protein expression in human acute myeloid cell lines

Please state whether Figure 2(b) summarizes the results of multiple biological experiments. It appears that the figure currently only reports the error within the FACS measurement of single samples. If it does not, please include multiple biological replicates to give readers an idea of the reproducibility of this diagnostic. I understand that the FACS plots are a single representative result, but at least one other element of the figure (e.g. the bar graph) should summarize the multiple biological replicates that are referred to in the main text.

One last minor point on Line 63 – Should be a reference to the authors’ previous Scientific Reports paper, i.e. ref 14. It currently appears to reference the wrong paper.

Author Response

Response to Reviewer 2 Comments

In this paper, the authors consider expression levels of KCTD15 on various types, building upon a previous Scientific Reports paper from 2019. It extends investigation of KCTD15 expression to lymphocytes, monocytes and granulocytes. It also finds KCTD15 expression is increased in several AML model cell lines. These findings are in line with the previously discovered role of KCTD15 in ALL. Overall, this paper represents an incremental advance over the previously published work. KCTD15 levels are found to be up-regulated in several new cell types, as well as several new oncogenic mutant cell lines. Its findings may be of interest to those in the field.

Scientifically, the paper requires better description and analysis of the results, as outlined specifically below. Otherwise, methods are well described and the results are clearly presented. No new experiments need be performed. The paper should be accepted once the problems below are addressed.

Response: We thank the reviewer for the supportive comments and for the constructive criticisms. Following the suggestions, we improved the results and discussion sections. In the current version of the manuscript, we (a) increased the number of peripheral blood samples studied, (b) performed a new experiment to discriminate HL60 cells when mixed in normal whole blood, (c) improved the significance of KCTD15 up-regulation in case of AML studying the KCTD15 mRNA expression using the open-access MILE study dataset.

Section 3.1. Intracellular KCTD15 expression in peripheral blood

Point 1: The purpose of the bar graph in Figure 1 is somewhat unclear – is the point that KCTD15 levels should be compare between different patients, or between cell types? If the latter, then perhaps plotting all four patients together would be more informative.

Response 1: Following the reviewer’s comment, we revised figure 1. The dot-plots from “a” to “e” refer to an exemplificative case extracted from the 14 studied. The panel “f” is a column graph showing the mean expression level of background, lymphocytes, monocytes and granulocytes studied in the cases included in this study. Each circle represents the measured value in that patient. The measured values are reported in Supplementary Table 1.

Point 2: In supplementary figures 2(e) and 3(e), the KCTD15 expression histogram displays a bimodal distribution for Lymphocytes. Please comment on why a large subset of Lymphocytes appears to have a much lower expression of KCTD15, but only in some patients.

Response 2: We thank for highlighting this aspect. We are not sure that the bimodal distribution found in normal lymphocytes is due to a biological phenomenon (caused, for example, by the presence of activated lymphocytes) or a technical bias due to primary or secondary antibody unspecific binding. Therefore, considering that these supplementary data could be confusing, we decided to remove the histograms and to present in table 2 the mean value and standard deviation recorded for each WBC subsets evaluated in our study population.

Point 3: Section 3.2. KCTD15 protein expression in human acute myeloid cell lines

Please state whether Figure 2(b) summarizes the results of multiple biological experiments. It appears that the figure currently only reports the error within the FACS measurement of single samples. If it does not, please include multiple biological replicates to give readers an idea of the reproducibility of this diagnostic. I understand that the FACS plots are a single representative result, but at least one other element of the figure (e.g. the bar graph) should summarize the multiple biological replicates that are referred to in the main text.

Response 3: We thank the reviewer for this consideration. We changed panel b of Figure 2 by inserting the correct graph containing the mean KCTD15 fluorescence derived from three independent experiments.

Point 4: One last minor point on Line 63 – Should be a reference to the authors’ previous Scientific Reports paper, i.e. ref 14. It currently appears to reference the wrong paper.

Response 4: We apologize for the mistake and thank the reviewer for pointing it out. We modified ref 14 with the correct reference (#11).

Reviewer 3 Report

First, there are a lot of English grammar errors in the authors' manuscript.    Extensive English editing for the manuscript is required.  

Second, it seems to be inappropriate to obtain a generalized conclusion of the increment of KCTD15 protein expression in acute myeloid leukemia based upon analysis of only two clinical patients or samples and three cell lines. Because the authors plan to use the high expression of KCTD15 protein as a biomarker in clinical diagnostics, such phenomenon should be proved in clinical cohort. 

In order to draw more generalized conclusion of the authors' analysis, authors can analyze and check the KCTD15 expression in the previously published available RNA-Seq and Proteome database of the clinical AML cohort.         

Author Response

Response to Reviewer 3 Comments

Point 1: First, there are a lot of English grammar errors in the authors' manuscript.    Extensive English editing for the manuscript is required. 

Response 1: We are sorry for the grammar errors present in the original text.  An extended linguistic revision of the paper was done upon revision.

Point 2: Second, it seems to be inappropriate to obtain a generalized conclusion of the increment of KCTD15 protein expression in acute myeloid leukemia based upon analysis of only two clinical patients or samples and three cell lines. Because the authors plan to use the high expression of KCTD15 protein as a biomarker in clinical diagnostics, such phenomenon should be proved in clinical cohort.

Response 2:Since we did not dispose of AML samples, to simulate them we mixed HL60 cells in normal peripheral blood, as described in figure 3 of the revised manuscript. The bright KCTD15 intensity of expression allowed us to discriminate HL60 cells from normal white blood cells. In addition, we included the MILE study dataset to demonstrate the KCTD15 up-regulation in AML samples as discussed below.

Point 3: In order to draw more generalized conclusion of the authors' analysis, authors can analyze and check the KCTD15 expression in the previously published available RNA-Seq and Proteome database of the clinical AML cohort.        

Response 3: Following this suggestion, we used the MILE study dataset to have a cohort of AML patients to be included in this study. Figure 4 displays the expression level of KCTD15 mRNA in four AML subtypes in comparison to healthy bone marrow. In all cases, the difference with healthy bone marrow was significant. We think that this finding significantly extend the implications of the manuscript and we thank the reviewer for the valuable suggestion.

Round 2

Reviewer 1 Report

The authors performed significant modifications by including additional samples and in silico datasets to support their findings. No further comments.